# Color-Indicating, Label-Free, Dye-Doped Liquid Crystal Organic-Polymer-Based-Bioinspired Sensor for Biomolecule Immunodetection

**DOI:** 10.3390/polym12102294

**Published:** 2020-10-07

**Authors:** Haw-Ming Huang, Er-Yuan Chuang, Fu-Lun Chen, Jia-De Lin, Yu-Cheng Hsiao

**Affiliations:** 1School of Dentistry, College of Oral Medicine, Taipei Medical University, Taipei 100, Taiwan; hhm@tmu.edu.tw; 2Graduate Institute of Biomedical Optomechatronics, College of Biomedical Engineering, Taipei 11031, Taiwan; 3Graduate Institute of Biomedical Materials and Tissue Engineering, and International PhD Program in Biomedical Engineering, College of Biomedical Engineering, Taipei Medical University, Taipei 100, Taiwan; eychuang@tmu.edu.tw; 4Division of Infectious Diseases, Department of Internal Medicine, Wan Fang Hospital, Taipei Medical University, Taipei 100, Taiwan; 96003@w.tmu.edu.tw; 5Department of Internal Medicine, School of Medicine, College of Medicine, Taipei Medical University, Taipei 100, Taiwan; 6Institute of Opto-Electronic Engineering, National Dong Hwa University, Hualien 97401, Taiwan; geman1218@yahoo.com.tw; 7International PhD Program for Biomedical Engineering, Taipei Medical University, Taipei 11031, Taiwan; 8Cell Physiology and Molecular Image Research Center, Wan Fang Hospital, Taipei Medical University, Taipei 100, Taiwan

**Keywords:** dye-doped liquid crystal, label-free biosensor, bovine serum albumin

## Abstract

The highly sensitive interfacial effects between liquid crystal (LC) and alignment layers make LC-bioinspired sensors an important technology. However, LC-bioinspired sensors are limited by quantification requiring a polarized microscope and expensive equipment, which makes it difficult to commercialize LC-bioinspired sensors. In this report, we first demonstrate that dye-doped LC (DDLC) chips coated with vertically aligned layers can be employed as a new LC-bioinspired sensing technology. The DDLC-bioinspired sensor was tested by detecting bovine serum albumin (BSA) and immunocomplexes of BSA pairs. The intensities of the dye color of the DDLC-bioinspired sensor can be changed with the concentrations of biomolecules and immunocomplexes. A detection limit of 0.5 µg/mL was shown for the color-indicating DDLC-bioinspired sensors. We also designed a new method to use the quantitative DDLC-bioinspired sensor with a smart-phone for potential of home test. The novel DDLC-bioinspired sensor is cheap, label-free, and easy to use, furthering the technology for home and field-based disease-related detection.

## 1. Introduction

The first liquid crystal (LC)-bioinspired sensor was invented in 2001 [1]. LC-based-bioinspired sensors have since become a hot topic of research [2,3,4]. The operating principle of the LC-bioinspired sensor is that biomolecules become immobilized on the aligned layers, leading to a vertical-to-planar response in the LC direction [4]. Consequently, the intensity of the transmitted light of the LC sensor changes, which can be observed under cross-polarized optical microscopy (cPOM) [5]. Today, LC-bioinspired sensors are used to detect many substances, such as cancer biomarkers, bovine serum albumin (BSA), immunoreactions, and DNA [1,3,4,6,7,8,9]. Among them, the most important is the detection of standard BSA for observation and establishing of basic characteristics [10,11,12]. However, the most important problem with LC-bioinspired sensors is the need for expensive equipment such as POM, electro-optical systems, and spectrometers. This makes it difficult to use LC-bioinspired sensors for home-based or clinical applications.

Cholesteric LC (CLC) is a special kind of LC material which has the property of Bragg reflection [13,14,15,16]. CLC-bioinspired sensors based on changing Bragg reflection have been proposed [17,18]. A sensitive color-indicating, label-free, easily used LC-bioinspired sensor has thus been demonstrated. However, CLC-bioinspired sensors still need expensive system equipment and complicated fabrication processes, which limits their application potential. There are many other label-free biosensing technologies. The most prominent of these techniques are plasmonic sensing and grating coupled interferometry (GCI) [19,20,21,22,23]. However, compared to GCI and plasmonic sensing, LC biosensors are cheaper and more portable, Furthermore, they can be measured with the naked eye [17,18].

In this study, we investigated dye-doped LC (DDLC) as a bioinspired sensing chip for quantifying certain important proteins: bovine serum albumin (BSA) and immunocomplexes formed of BSA pairs. BSA from polypeptide samples is an essential standard of calibration, and a model for the use of dichroic pigments in the liquid crystal industry. The light absorption of these dyes can be controlled by adjusting the orientation of the LCs. This technique produces reflective guest-host LC displays without backlighting.

The absorption of DDLCs at 420 nm was used for the spectrometric analysis of concentrations of BSA and its immunocomplexes. By using vertically aligned DDLCs and incorporated by the alignment layer amphiphilic agent, DMOAP, 1-octadecanaminium, and *N*,*N*-dimethyl-*N*-[3-(trimethoxysilyl)propyl] chloride. The DMOAP was often used for homeotropic anchoring in nematic LC environment. The disruption of the immobilized DDLCs with various concentrations of BSA antigen/antibody pairs was examined. The results demonstrated the sensitivity of the DDLC-based-bioinspired sensing chip for colorimetric immunodetection. Furthermore, our research team employed a smart-phone with imaging software functionality to quantify BSA concentrations with a lower detecting limitation of around 0.5 ng/mL. Our team thus created a promising smart-phone-based DDLC device capable of being used for measuring BSA concentrations with a system of achromatic microscopy aimed at the smart-phone, and showed that acquiring images from smart-phone was achievable.

## 2. Materials and Methods

### 2.1. Materials

The organic-polymer based sensor were created by doping with red dichroic dyes (DR1, Sigma-Aldrich, St. Louis, MO, USA) in powder form at a concentration of 2 wt% into the eutectic nematic host E7 (Merck). The phase transition temperature of the DDLCs was 94 °C. As shown in Figure 1, the absorbance of DDLCs was measured using a spectrophotometric approach. Absorption spectra of the DDLC were acquired with an ultraviolet-visible (UV-Vis) spectrometer (Shimadzu UV2600, Shimadzu Corporation, kyoto, Japan). The absorption bands of DDLC with a maximum value at 420 nm were observed using to the chromophore compound as shown in Figure 1. The glass substrates were immersed in the DMOAP fluid-form solution for 0.5 h in order to coat a layer of appropriately aligned shape. Then, we rinsed the substrates with deionized (DI) water for approximately 1 min to remove extra DMOAP. We use a heating platform to dry excess water from substrates. To test the immobilization of BSA, we applied an aqueous solution with 0.2 mL of BSA to the alignment layer of DMOAP-dipped glass substrate. BSA concentrations of 0.5–10 µg/mL were employed. Various anti-BSA antibody concentrations were used as well (0, 10, 100, and 1000 µg/mL). BSA protein (Sigma-Aldrich, St. Louis, MO, USA) is the most common protein standard for such measurements, with 583 amino acids and a molecular weight of 66.43 kDa. BSA’s molecular weight is close to the mean molecular weight of human proteins (around 40–50 kDa). Therefore, if a DDLC-bioinspired sensor can detect BSA, it should be applicable to many other relevant proteins.

### 2.2. Fabrication Methods

Figure 2 demonstrates the steps for fabricating a DDLC-bioinspired sensor. To prepare a DDLC device for detecting BSA and immunocomplexes of BSA pairs, we immersed indium-tin oxide-covered glass substrates for 0.5 h in a fluid-form solution, including 1.6% (*v*/*v*) of DMOAP in DI water to produce DMOAP-coated substrates at ambient temperature (Figure 2b). Then we rubbed the substrates using a rayon-clothing rubbing machine at 1500 rpm, as shown in Figure 2c). Through rubbing of the aligned layers, alignment defects can be reduced, suppressing background noise during detection. In other words, rubbing improves the detection limit. Prior to the BSA immobilization experiment, a 1, 0.5, 1, 2, 4, 6, 8, or 10 μg/mL BSA solution was dispersed onto a DMOAP substrate for 30 min (Figure 2d). Once rinsed with DI water to remove unbound BSA, cells of each DDLC were self-assembled with substrates of 2-DMOAP-coating with a 10-μm spacer used to make a cell gap (Figure 2e). Then, the different BSA concentrations were used to fill the DDLC cells by capillary action (Figure 2f). Then, in the second experiment, an anti-BSA antibody was immobilized on the aligned layer. Figure 2g shows that the BSA antigen reacted with the anti-BSA antibody, forming an immunocomplex of the BSA antibody/antigen pair. Finally, each substrate was assembled with 10-μm spacers to form a sandwich-structured device (Figure 2h). To examine the electro-optical properties, a probe laser beam derived from a laser system was used. An arbitrary function generator (Tektronix AFG-3022B, Tektronix, Inc., Beaverton, OR, USA) was used to apply voltages.

## 3. Results and Discussions

### 3.1. Mechanisms of the DDLC Biosensors

The given dichroic characteristics and absorbance of DDLCs gives DDLC-bioinspired sensors a color-indicating property. In our experiment, a DDLC-bioinspired sensor was placed on white paper, effectively exposing it to unpolarized white light normally incident on the sensor (shown in Figure 3). Because the maximum absorption of the DDLCs was at 420 nm, an output light exhibited the corresponding red. Absorption of the anisotropic dichroic dye by the LC macromolecules was influenced by the pretilt angle with respect to the polarization of incident light and LC molecular axis. Polarization parallel to the long axis of the DDLC macromolecules maximized the absorbing intensity happened. The smallest absorbing intensity was encountered when the direction of polarization was perpendicular to the long axis of the macromolecules. However, without BSA, a weak absorbing intensity was shown in the direction of polarization when unpolarized white light entered perpendicular to the long axis of the DDLC molecules, parallel to the plane of the cell (Figure 3a). Once we immobilized BSA onto the DMOAP-coated glass, the vertical alignment power of the DDLCs was disrupted, and the DDLC macromolecules sloped away from their orientation normal to the plane of the substrate. This caused an increase in the parallel component of the DDLC direction with respect to the polarization direction (Figure 3b). The transmittance at 420 nm declined as the BSA concentration increased. Because of the enhanced dye absorption, the DDLC-bioinspired sensor’s color strengthened in the presence of BSA.

### 3.2. Smart-Phone Images from DDLC Biosensors

As shown in Figure 4, to examine this, we placed a smart-phone onto a phone stage (From Scimage). We aligned the phone camera with an arrangement of magnifying lenses in the phone stage. Opposite the set of magnifying lenses, we designed a simple sample chip tray for holding the device. The sample was sandwiched between two 130-µm thick cover glasses of dimensions 22 mm × 22 mm, separated by a parafilm spacer. A set of achromatized lenses containing dual-convex lenses with concave-plane lenses was used for imaging texture of the DDLC, to reduce chromatic and spherical deviations. Using the designed apparatus, the DDLCs can be graphically imaged with the achromatic lenses, with ~5× magnification. The DDLC’s photograph sample was taken by the smart-phone, as demonstrated in Figure 4c. The color of the DDLCs can be measured by the smart-phone’s camera, as displayed in Figure 4d. In this new design system, the biosensor was imaged using a 5× magnification achromatic lens with a diameter of 10 mm and a focal length of 4 mm, as confirmed by using a calibration ruler. As shown in Figure 5, the bright white state of DDLCs under the smart-phone camera without BSA was induced from the homotopic state of the DDLCs by the DMOAP alignment layer. When BSA was immobilized onto the surface of the DMOAP coating, the anchoring power of DMOAP seemed to be shielded. The DDLC direction was twisted from the normal plane of the substrate, producing a color transition of the optical texture of DDLC. The gradual increase in red color saturation of the optical states of DDLCs at BSA concentrations ranging from 1 to 10 μg/mL is shown in Figure 5.

### 3.3. Quantitation and for DDLC Biosensors

In addition, the DDLC texture from the smart-phone was adopted to allow quantification by imaging software (image J) with a quantitation brightness function [16]. The approximately linear relationship between the optical density from software quantitation pixels brightness process of the DDLC chip and various concentrations of BSA are shown in Figure 6. Increasing concentrations of BSA yielded lower optical densities. An approximately logarithmic scale was used to permit detection of biomolecules with the designed smart-phone-based system. The calibration standards for different major brands of smart-phones are proposed (Figure 6) to make the system available for various brands of smart-phone. Information on detectable behaviors was measured using transmission spectra based on the absorption properties of the DDLCs. Two wavelength regions of the absorption regime (λ = 360–500 nm) and the non-absorbing range (λ = 500–600 nm) are shown in Figure 7. A valley of optical transmission, including a minimum central intensity at approximately 420 nm, was shown in the DDLCs’ band. In DDLC cells with a 10 μm cell-gap, transmittance at λ = 420 nm decreased as BSA concentrations rose (Figure 7a). At 10 μg/mL of BSA, the spectrum of transmission of DDLCs was lower than that in the presence of 8 mg/mL of BSA. We conclude that minimum transmittance occurs at λ= 420 nm, and that differences in concentration of BSA could not be detected less than 8 μg/mL. To generate a BSA calibration curve for quantification of protein amounts, the standard parameter *S%* was derived to indicate the related loss of optical transmittance signal wavelength at λ = 420 nm for BSA concentrations. This parameter was calculated with the equation below [24]:$$S\;\%=\left(\frac{T_{0}\;-\;T_{BSA}}{T_{0}}\right)\times 100\%;$$
where *T*_0_ and *T*_BSA_ are defined as the optical transmittance at λ = 420 nm of a DDLC without or with BSA, respectively. Through calculating *S%* values analyzed from Figure 7a against the concentration of BSA, a linear relationship with a determination coefficient R^2^ ≥ 0.93 was found between 0.5 and 10 mg/mL of BSA (Figure 7b). In addition to the optical anisotropic dichroic features, the electrical properties induced by the electro-optical features of DDLCs should be a helpful tool for-bioinspired quantitative sensing. To investigate the voltage-dependent optical properties, an arbitrary function generator was used to supply square-wave voltages from 0 V to 10 V at 1 kHz across the ITO of DDLC cell to induce the orientation of LC through dielectric coupling with the electric field. As displayed in Figure 7a, the voltage-dependent optical transmittance wavelength at λ = 420 nm was investigated under external voltage fields of 0–10 V_rms_, which were applied to the device at various immobilized BSA concentrations.

### 3.4. Quantitation for Immunoassay DDLC Biosensors

The positive correlation between *S*% and concentrations of BSA of 0.5 and 10 μg/mL is illustrated in Figure 8b. Our results demonstrated that both the dielectric and optical anisotropic characteristics of DDLCs can be used to establish a precise DDLC-constructed sensing method for quantification of bio-macromolecules. To make our DDLC-bioinspired sensors more suitable for medical use, the immunoassay performance of the sensors was assessed using BSA antibodies. The immunoassay light transmittance intensities test, which immobilized by 0–10 μg/mL BSA concentrations and 0 to 1000 μg/mL anti-BSA antibody concentrations, yielded results shown in Figure 9. The immune complexes associated with BSA antibodies with different BSA antigen concentrations are also shown.

We see that at low anti-BSA antibody concentrations of <10 μg/mL, BSA immune complexes cannot organize the pairs of antibody/antigen. The optical densities did not clearly increase for BSA of 1 or 10 μg/mL. However, an excess anti-BSA antibody concentration significantly affected the arrangement of DDLC, making the light intensity of different BSA immune complexes less clearly identifiable. The quantitative results for the immunoassay experiment show that BSA immune complexes can be used in the DDLC system. Anti-BSA antibody concentrations of 1 μg/mL were most suitable in the DDLC device, based on the data. Thus, a DDLC-bioinspired sensor for immunodetection is successfully demonstrated herein. The fact that BSA can be quantified by the DDLC biosensor suggests that other proteins can also bind in immunoassays, or nonspecifically, to DDLC, and be quantified in a similar fashion. At present, various protein quantification methods have been developed and assessed for their sensitivity [25,26,27]. The detection limits of several absorption- and fluorescence-based protein assays have been shown in past papers [24]. The BSA concentration range detected by the DDLC-based biosensor was about 0.5 μg/mL, which is the same order as that measured by absorbance-based protein sensors [28]. This suggests that the DDLC biosensor may be used to estimate the concentration of most protein samples. However, the detection limit of DDLC biosensors can be improved by optimizing the birefringence and dichroic ratios of DDLC [29]. Compared with the past DDLC research [27,30], we use cheaper and common materials to fabricate the device and can be observed by using smart-phones without using polarizers and microscopy, which may greatly increase the possibility of commercialization and home-use.

## 4. Conclusions

The highly sensitive interfacial effects of DDLC-bioinspired sensors are discussed here, and the creation process of such as sensor is outlined. The sensor was tested by using it to detect BSA and immunocomplexes of BSA pairs, and these results were discussed in this report. Concentrations of biomolecules and immunocomplexes both affected the intensities of the dye color of the DDLC sensors. A limit of detection (0.5 µg/mL) was exhibited for color-indicating DDLC sensors. In addition, a new method using a smart-phone to quantify biomolecules in the DDLC sensor was successfully invented. This color-indicating DDLC technique can enhance development of LC-bioinspired sensors as biomarkers for diagnosing disease in the home, in clinical applications, and in microfluidic immunoassays [31]. Based on the immunoassay results of DDLC, we may be able to change to use anti-SARS-CoV-2 antibodies to capture SARS-CoV-2 and affect the arrangement of DDLC. Combined with smart-phone detection, this may have great advantages for coronavirus disease quarantine.

## Figures and Tables

**Figure 1 polymers-12-02294-f001:**
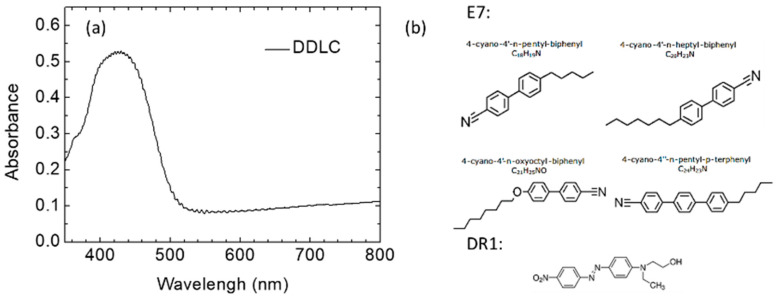
(**a**) The absorption spectra of the DDLC device. (**b**) Chemical structure of the dye-doped liquid crystal (DDLC) compound.

**Figure 2 polymers-12-02294-f002:**
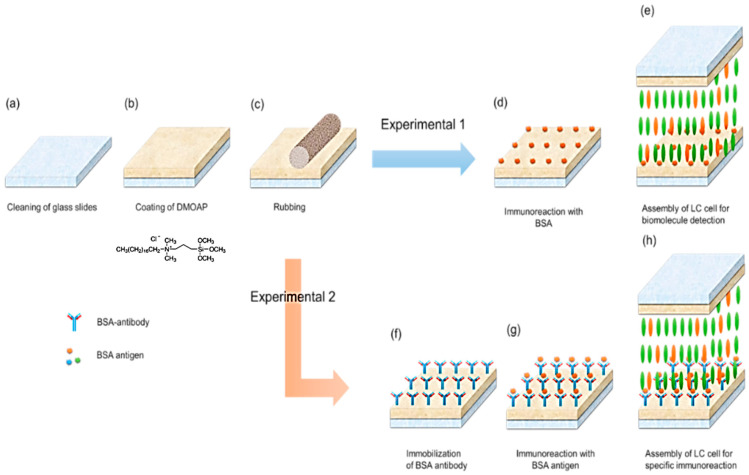
Schematic of the sample preparation: Experiment 1: (**a**) cleaning the substrates, (**b**) preparing alignment layer DMOAP-coated substrates, (**c**) rubbing the DMOAP-coated substrates unidirectionally, (**d**) dispensing bovine serum albumin (BSA) on one of the substrates, and (**e**) bestowing the spacer on individual parts near the substrate’s long edges, followed by self-assembling two coated substrates to form an empty device and introducing DDLC to the cell. Experiment 2: (**f**) dispensing the BSA antibody on one of the substrates, (**g**) dispensing the BSA antigen on the substrate with the BSA antibody, (**h**) self-assembling the two coated-substrates to form a sandwiched empty device, and introducing DDLCs into the device.

**Figure 3 polymers-12-02294-f003:**
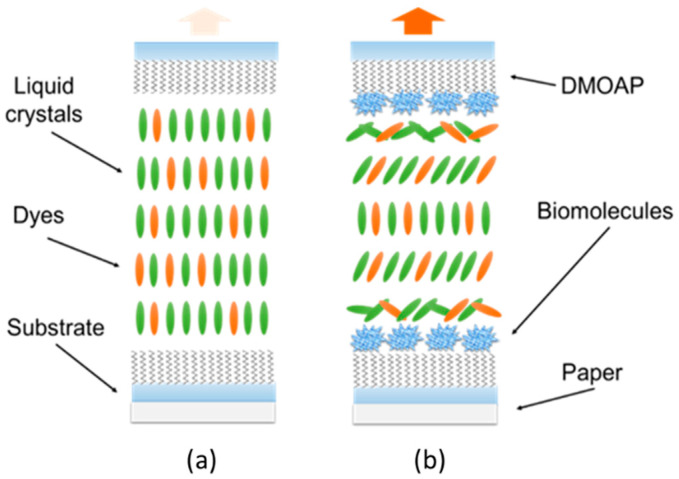
Schematic illustration of the DDLC-based-bioinspired sensing platform in the absence of, and immobilized with biomolecules. (**a**) without biomolecules and (**b**) with biomolecules.

**Figure 4 polymers-12-02294-f004:**
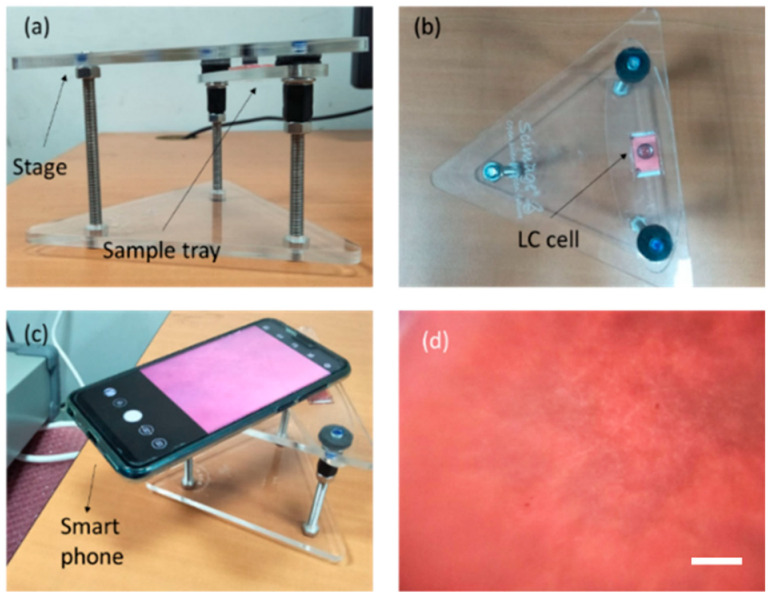
(**a**–**c**) Smart-phone accompanying designed development for a DDLC sensor chip. (**d**) DDLC macromolecules from the smart-phone’s camera. The scale bar is 300 μm.

**Figure 5 polymers-12-02294-f005:**
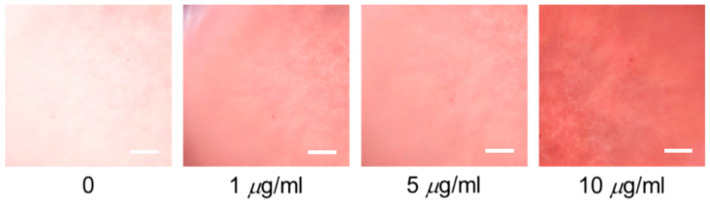
Optical textures of the DDLC device in the presence of BSA concentrations of 0 to 10 μg/mL. The scale bar is 300 μm.

**Figure 6 polymers-12-02294-f006:**
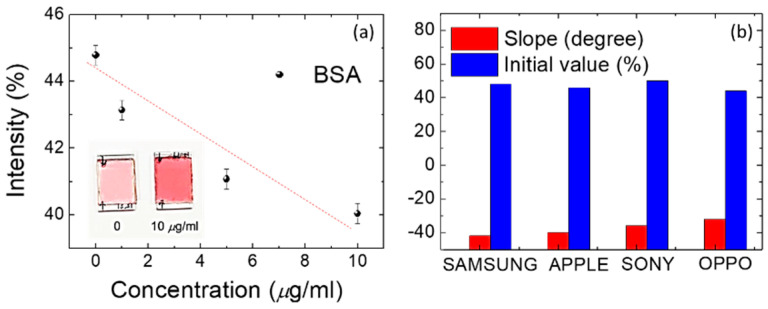
(**a**) The relationships of the light intensities from image integration software of a DDLC-bioinspired sensor exposed to several BSA concentrations. (**b**) The calibration standard of different major brand smart-phones.

**Figure 7 polymers-12-02294-f007:**
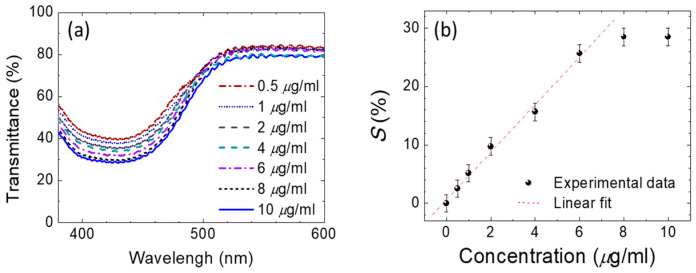
(**a**) Transmission spectra of DDLC cells at bovine serum albumin (BSA) concentrations ranging from 0.5–10 μg/mL. Correlation between the parameter of standard *S%* (the wavelength transmittance-related loss) and BSA concentrations in DDLC cells. (**b**) Linear relationship with a determination coefficient R2 ≥ 0.93 was found between 0.5 and 10 mg/mL of BSA.

**Figure 8 polymers-12-02294-f008:**
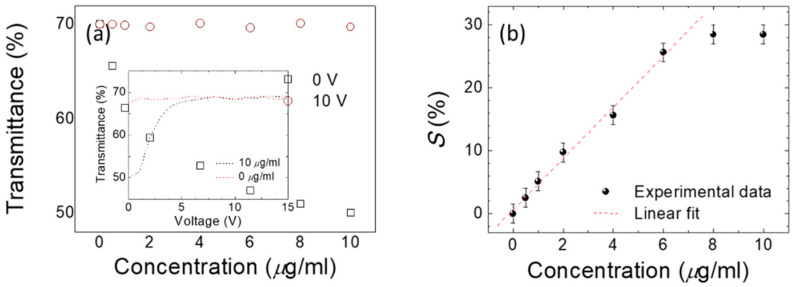
(**a**) Intensity of DDLCs immobilized by 0–10 μg/mL BSA concentrations under 0 and 10 V. Inset, voltage-dependent transmission spectra at DDLC cells immobilized with 0 and 10 µg/mL of BSA. (**b**) Relationship between the parameter of standard *S*% (the wavelength transmittance-related loss) and BSA concentrations in DDLC cells.

**Figure 9 polymers-12-02294-f009:**
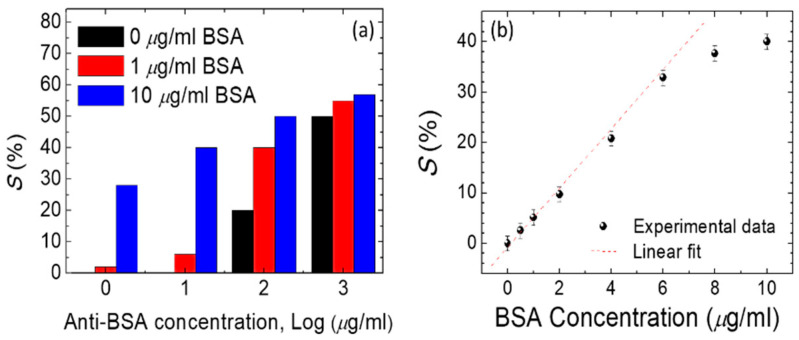
(**a**) Intensities of immunoassay DDLCs immobilized with concentrations of BSA of 0–10 μg/mL along with concentrations of anti-BSA antibody of 0, 10, 100, and 1000 μg/mL, (**b**) Relationship in the parameter of standard *S*% (the wavelength transmittance-related loss) and BSA concentrations at an anti-BSA concentration of 1 µg/mL.

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
