# Peer review of "Color-Indicating, Label-Free, Dye-Doped Liquid Crystal Organic-Polymer-Based-Bioinspired Sensor for Biomolecule Immunodetection"

_polymers, 2020, doi:10.3390/polym12102294_

Round 1

Reviewer 1 Report

The work is interesting and the manuscript is well written. However, these minor modifications related to organization would be suitable.

  1. Section 2 should be divided to clearly demonstrate the materials and methods used.
  2. Figs. 1 and 2 should come with Section 2, not in Section 3. Then, the other figures should be in line with the discussion.
  3. Figure 6 should be a and b. The quality of Figure 6 (left) is very low and need improvement.
  4. The inset of Fig. 8a is not clear.

Reviewer 2 Report

Manuscript ID: polymers-945077
Title: Color-indicating, label-free, dye-doped liquid crystal organic-polymer based bioinspired sensor for biomolecule immunodetection
Comments: In this manuscript, authors report a liquid crystal (LC)-based device that can sense a presense of biomolecules. The mechanism is based on light transmittance in a device filled with LC between two polarized plates. When one of the polarized plate is exposed to biomolecules, they can disturb the alignment of LC, resulting in a change in the transmittance. Detection can be made using a smartphone camera. In my opinion, this manuscript requires revision before consideration of publication in this journal. First of all, I do not see any significant findings or improvement from this work. LC-based sensors have been fabricated before. BSA has previously been the target compound, too. In addition, I do not see how this work is of the topic of the journal Polymers. In the title, authors specified "organic-polymer based" but there is no novel and active functions of any polymeric layer. There are some other issues regarding this manuscript, and should be addressed and resolved before submission:
(1) The novelty and performance of this work, compared with the other previous reports in literature, should be stated in this manuscript.
(2) Authors should present the chemical strcutures of the materials (dye, LC, sensing targets, antibody, etc.) used in this work.
(3) What is DMOAP? This should be clearly defined and explained in the manuscript.
(4) For certain experiments, voltage was applied. However, it is not clear how and why the voltage is applied in the system. This should be clearly stated in the manuscript.
(5) The quality of figures are quite poor. Especially, Figures 2, 5, 6, 7, 8, and 9 should be presented in a better quality.
(6) Figure 9(a): How is %S not zero for the sample with 0 ug/mL BSA?
(7) There are some typos, for example, "senor were" in line 80. Please carefully read through the manuscript and correct typos.
